# Research on the Vitality Evaluation of Parks and Squares in Medium-Sized Chinese Cities from the Perspective of Urban Functional Areas

**DOI:** 10.3390/ijerph192215238

**Published:** 2022-11-18

**Authors:** Tongwen Wang, Ya Li, Haidong Li, Shuaijun Chen, Hongkai Li, Yunxing Zhang

**Affiliations:** School of Architecture and Artistic Design, Henan Polytechnic University, Jiaozuo 454000, China

**Keywords:** parks and squares, vitality, medium-sized cities, urban functional areas

## Abstract

Medium-sized cities are increasingly committed to the planning and construction of urban public spaces to meet people’s demand for high-quality urban life. Parks and squares are important parts of urban public spaces, and their vitality represents the quality of public spaces to a certain extent and reflects the happiness index of urban residents. At present, the functional areas and transportation networks of medium-sized cities are still developing. Due to the influence of urban construction, the planning of parks and squares in medium-sized cities has not yet caught up to that in larger cities. This study analyzed a medium-sized city, Jiaozuo, as an example, with the help of point of interest (POI) data, OpenStreetMap road network density data and WorldPop population data. The vitality of parks and squares in different functional spaces in the main urban area in Jiaozuo was quantitatively evaluated in terms of the four following aspects: urban space functional area characteristics, travel vitality index of urban residents, park and square attractiveness and the regional service levels of parks and squares. The effects of functional mixing, traffic network density, population density and spatial distribution on the vitality of parks and squares in medium-sized cities were also studied. The results showed that (1) the functional mixing in the main urban area in Jiaozuo was characterized by a spatial distribution of high in the center and low in the surrounding areas, with the highest functional mixing in the central part of the Jiefang District; (2) the travel dynamics of urban residents were characterized by a clear development of concentric circles radiating in a circular pattern; (3) the levels of service in parks and squares were particularly high in Jiefang District, with a spatial distribution of Jiefang District > Shanyang District > Macun District > Zhongzhan District; (4) under the condition that the service levels of each district were the same, the vitality values of the existing parks and squares in each district were compared and, from high to low, were Jiefang District (1.0–3.5), Shanyang District (0.2–2.0), Macun District (0–1.4) and Zhongzhan District (0–1.2). Functional mixing, road networks and population density had significant impacts on the vitality of parks and squares. Based on our study on the division of urban functional areas, we expanded the study to include urban microspaces. By evaluating the vitality of existing parks and squares and analyzing the influencing factors of spatial vitality, we found that it would be helpful to adopt targeted strategies to improve spatial vitality. Considering the spatial layouts of parks and squares, planning and constructing high-vitality parks and squares would be conducive to the future development of medium-sized cities. The existence of high-vitality spaces could also help to realize the sustainable development of cities.

## 1. Introduction

The vitality of leisure spaces is an important manifestation of urban vitality. As important parts of urban leisure spaces, parks and squares are important recreational places for urban residents [1,2] and provide cultural ecosystem services, including spiritual enrichment, cognitive development, a sense of place and recreational experiences [3]. Parks and squares can host human activities and attract people to participate, thereby bringing vitality to urban spaces and reflecting the levels of vitality in urban spaces. With improvements in the living standards of urban residents, the demand for parks and squares in large cities has continued to rise, and the leisure and recreational functions of parks and squares in medium-sized cities have also attracted much attention. Citizens not only pay attention to the quantity and quality of parks and squares but also to the convenience of accessing parks and squares [4,5]. The focus of public attention is an important factor that affects the vitality of parks and squares. The number, quality and accessibility of parks and squares in cities affect their utilization rate to a certain extent. A high public space utilization rate is conducive to improvements in urban vitality. As an important factor of urban space vitality [6], the vitality of parks and squares also reflects the happiness index of urban residents. In recent years, with the rapid development of urbanization, construction land in cities of different scales has been expanding, and land structures have been frequently adjusted. In addition, differences in road network density and population distributions have led to different levels of construction of parks and squares in cities. Furthermore, some gaps remain between the construction levels of urban parks and squares at different levels. For large cities with higher economic levels, such as Beijing and Shanghai, the urbanization rate has reached its peak, and most public spaces, such as parks and squares, have shifted to the process of stock improvement to gradually meet the needs of residents at a higher level [7]. Urban construction is at a high level, such as improvements in traffic networks, functional mixing, public service facilities, etc. However, parks and squares in medium-sized cities are still in the stage of incremental development [8]. Compared to the high-vitality parks and squares in large cities, the vitality of parks and squares in medium-sized cities is relatively low. By analyzing the differences in construction levels, traffic network density and functional mixing degrees between large cities and medium-sized cities, the number of parks and squares and spatial layout maturity can be extracted as factors that affect the spatial vitality of parks and squares [9,10]. Of course, there is another factor that cannot be ignored, namely, the urban population. City size can be categorized according to the number of permanent urban residents. As the main user of space, population also affects the use of parks and squares [11]. By taking the above influencing factors as the evaluation indices of vitality, a vitality evaluation framework was determined in this study, and measures to improve the vitality of parks and squares in medium-sized cities were proposed.

Existing research on spatial vitality can mainly be divided into qualitative and quantitative research according to the research methods used. Qualitative research mainly analyzes data obtained through interviews and questionnaires [12,13], which are characterized by strong subjectivity [14], whereas quantitative research mainly analyzes the quantitative characteristics, relationships and trends of change in large amounts of data [15]. With the advent of the era of big data, research data sources have expanded from traditional channels, such as questionnaires, to network channels, such as POI data. Research methods have also expanded from quantitative analyses to ArcGIS spatial analyses [16,17]. Compared to traditional data, big data has the advantages of easy access and dynamic updating and contains sufficient physical and objective information to describe urban spatial environments [18]. For example, the spatial accuracy of mobile applications that provide location services via GPS is approximately 10 m [19]. This method has been used to study the vitality of various urban spaces and has verified their feasibility, scientificity and rationality. For example, multisource big data has been used to quantitatively assess the vitality of the underground spaces of urban metros [20]. Based on multisource geospatial big data, 11 indicators were selected from the three dimensions of population, land use and transportation to model the influence mechanisms of spatial structures on urban vitality [21]. Spatial and temporal changes in urban vitality were quantitatively measured using Baidu heatmaps, and the influence of building environmental factors on urban vitality was analyzed using a geographically weighted regression model [22].

Research into space vitality at home and abroad has mainly been reflected in the following three aspects: (1) factors affecting space vitality, i.e., the influence of building environments on spatial vitality can be analyzed by the factors of street space networks, functional density, diversity and accessibility [23,24]; (2) the evaluation and measurement of spatial vitality, i.e., some scholars have selected static environmental indicators to quantify spatial vitality (such as infrastructure density [25], functional mixing [26], commercial facility density [27], green landscape components [28] and accessibility and livability [29]), whereas other scholars have measured spatial vitality in terms of the two dimensions of time and space (such as correlations between the spatial and temporal distributions of vitality zones and land use [30], vitality intensity and stability [31] and the use of Tencent real-time user density data to measure the regularity of park visits [32]); and (3) the creation of space vitality, i.e., spatial vitality can be effectively promoted by increasing functional areas around the space and improving the surrounding road networks [33,34].

The planning and management of parks and squares implement “people-oriented” design concepts [35]; their vitality is closely related to public participation, and people’s use of parks and squares is largely affected by their attributes [36], for example, park size, park age [30], aesthetic characteristics [37], internal spaces and facilities [38,39]. To a certain extent, park attributes represent what attracts urban residents to parks [40], and better internal service facilities can attract more users [41]. In most studies, the distance between users and a space has been one of the most important factors affecting the use of parks and squares [42,43]. Studies have shown that for parks and squares that are farther away, people have fewer opportunities to participate in activities [44,45]. In addition, the accessibility and the surrounding environment of spaces also affect the motivation of urban residents to go to parks and squares [37,46]. In summary, with the innovation in relevant research methods, research into spatial vitality using multisource data has gradually increased, although the evaluation of factors that affect spatial vitality is still in the exploratory stage. At present, research into the vitality of urban parks and squares at home and abroad has mostly been based on other existing research, and the evaluation indicators have been more focused on selecting the elements within spaces, such as space design and space care [47]. The surrounding environments of parks and squares also have significant impacts on the vitality of spaces. Therefore, this study focused on the spatial distribution of parks and squares in medium-sized cities. Because the definitions and classifications of parks and squares have not been clearly established, the parks and squares studied in this paper were urban areas greater than 0.2 hm^2^ in scale. The parks and squares were considered as independent research units. Based on the existing research on the division of urban functional areas [48,49], an analysis of urban microspaces was also added. Taking Jiaozuo as an example, the influence of material and environmental factors on the vitality of parks and squares was analyzed according to urban functional areas, and a vitality index was determined. The vitality indicators were selected according to the following two aspects. First, by analyzing the construction levels of large cities and medium-sized cities, it was found that there are large gaps in terms of functional layout, traffic network density, the number of parks and squares and their spatial distribution [8,9]. Second, according to the literature, the material environmental factors that affect the vitality of public spaces include urban spatial function, traffic convenience, surrounding development intensity, distance to destination, accessibility, space design and more [33,50,51,52,53,54]. Considering the availability of data and the requirements for the quantitative calculation of the indicators, this study established the four dimensions of urban functional mixing, spatial population distribution, traffic network density and attractiveness of parks and squares as a vitality impact factor index system for urban parks and squares.

By comparing the gaps in the construction of parks and squares in cities of different sizes in China, this study differed from previous studies related to urban spatial vitality (evaluating spatial vitality from the perspective of internal spatial factors). The factors affecting the vitality of parks and squares in medium-sized cities, particularly from the perspective of external factors affecting spatial vitality, were analyzed in this paper. The relationships between the mix of urban functions, spatial distribution of population, density of road network, attractiveness of parks and squares and their levels of service on the vitality of parks and squares in medium-sized cities were explored. An evaluation system for urban spatial vitality was constructed by comparing the high and low vitality of parks and squares in different areas of the studied city. Strategies for activating spatial vitality were proposed for different construction levels of regions, which were conducive to improving the spatial vitality of parks and squares in each district. This paper also provided suggestions for medium-sized cities to plan and build public spaces in the future and promoted the formation of more highly dynamic spaces. We believe that this study could contribute to evaluating urban spatial vitality and achieving sustainable spatial development in medium-sized cities.

The remainder of the paper is structured as follows. Section 2 introduces the study area and multisource data. Section 3 details our proposed approach for vitality evaluations of parks and squares. Section 4 presents the results of the comparative evaluation in the study area. Finally, Section 5 summarizes the main conclusions of this work and offers directions for future research.

## 2. Overview of the Study Area and Data Sources

### 2.1. Study Area

As shown in Figure 1, the main urban area in Jiaozuo comprises four districts covering a total area of 489.81 km^2^, which accounts for 12.03% of the total area of the city. The resident population in this area numbers 878,500 people, accounting for 24.95% of the city’s total population. As the older urban areas in Jiaozuo, the Jiefang District and Shanyang District are densely populated and rich in industry. With the construction of modern cities, how to stimulate the vitality of urban spaces and realize refined management and development while protecting the historical features of the old cities has become a difficult challenge.

### 2.2. Sources of Data

POI data: The 2020 Gaode map data were obtained using crawler technology. The data content included basic information, such as the category, address, name, latitude and longitude of each point of interest, for a total of 41,961 data points. After filtering out duplicate and incorrectly addressed data points, a total of 41,902 valid points were retained (Figure 2), including 238 parks and squares within the study area. To ensure the consistency and universality of the data, all POI data were classified into 13 categories and several subcategories according to standard POI classification (Table 1).

Population data: The methods for obtaining the spatial population distribution data included WorldPop, GPW v4 and two Chinese kilometer grid population distribution datasets. We selected WorldPop, as it has the highest overall accuracy. The WorldPop dataset has a spatial resolution of 1000 m. It is a high-resolution population distribution grid map based on nighttime light data, land use data and other information. The weight layer of the population distribution is estimated using a random forest model. Based on the WorldPop China 2020 population dataset, the population data for each spatial grid unit in the main urban area in Jiaozuo were obtained via mask extraction, grid turning points and spatial connections using the ArcGIS platform (Figure 3).

Road network data for the main urban area in Jiaozuo: OpenStreetMap road network data presently offer some of the most widely used and representative volunteer geographic information projects. OpenStreetMap aims to establish a free and open map data source. The data type is a vector format, which met our research requirements. A total of 1216 road network data points were obtained (Table 2).

## 3. Research Methods

### 3.1. Research Framework

A vitality evaluation system for parks and squares in medium-sized cities was established using four dimensions: urban functional mixing, the travel vitality index of urban residents (including spatial population distribution and traffic network density), the service levels of parks and squares and the attractiveness of parks and squares. We combined the relationships between these various dimensions and used multisource data to determine the research ideas for this article (Figure 4). By cleaning and classifying the POI data, the basic discrimination of urban functional areas and the locations of parks and squares in Jiaozuo’s main urban area were obtained. The single functional and mixed functional areas in Jiaozuo’s main urban area were quantitatively identified using the partition identification method. A significance model for the POIs was constructed in terms of public awareness, and the travel vitality index of urban residents was calculated by integrating WorldPop population density data and OSM road network density data. We classified the parks and squares and assigned them levels of attractiveness. Finally, combined with the two indicators of service coverage and service overlap rate, the overall vitality of parks and squares was comprehensively measured, and the spatial distribution and vitality of parks and squares in the four districts in Jiaozuo’s main urban area were comprehensively evaluated.

### 3.2. Quantitative Identification of Functional Areas

As a research hot spot within urban science [55], the identification of urban functional areas was one of the important research processes in our method. To ensure the consistency of POI, population and other data with geographic information, we followed the spatial resolution of the WorldPop population density dataset and used a square grid of 1000 m × 1000 m as the basic research unit. There were 594 functional area units in the study area. We used the partition identification method to identify functional properties by constructing the frequency density ($F_{i}$) and type proportion ($C_{i}$). The formulae were as follows:(1)$$F_{i}=\frac{n_{i}}{N_{i}}\left(i=1,2,\ldots ,13\right)$$
(2)$$C_{i}=\frac{F_{i}}{\sum_{1}^{13}F_{i}}\times 100\%\left(i=1,2,\ldots ,13\right)$$
where i denotes the POI type, $n_{i}$ denotes the number of POIs of type i in the cell, $N_{i}$ denotes the total number of POIs of type i, $F_{i}$ denotes the frequency density of the POIs of type i to the total number of POIs of that type, and $C_{i}$ denotes the ratio of the frequency density of the POIs of type i to the frequency density of all POIs of that type in the cell.

Using Equations (1) and (2), the frequency density and type ratio of each unit were calculated, and a type ratio value of 50% was determined to be the standard against which to judge the functional properties of the units. When the type ratio $C_{i}$ of a certain type of POI data in a unit sample was 50% or more, its functional nature was determined by the type of POI data, which was a single functional area of this type. When the $C_{i}$ value of all POI types in the sample was less than 50%, their functional properties were determined by the two types of POI data types that accounted for the largest proportions in the unit sample. When there were no POI data in a sample, it was considered as a data-free area.

### 3.3. Travel Vitality Index of Urban Residents

Different functional areas have different attractions for urban residents. The population and road network densities in the spatial grid units were used in the travel vitality index model to normalize and superimpose the data. Then, the weight coefficients of the different functional areas were multiplied to obtain a more accurate spatial distribution of population travel vitality in urban functional areas. The formulae were as follows:(3)$$G_{xm}=\frac{G_{m}-G_{x}^{min}}{G_{x}^{max}-G_{x}^{min}}$$
(4)$$C_{m}=G_{xm}\times Q_{y}\left(x=1,2;y=1,2,\ldots ,13;m=1,2,\ldots ,n\right)$$
where x = 1 and x = 2 denote the population and road network densities, respectively, m denotes the number of functional areas, $G_{m}$ denotes the value of cell m, $G_{x}^{max}$ denotes the maximum value of class x, $G_{x}^{min}$ denotes the minimum value of class x, $G_{xm}$ denotes the normalized value of class x for the functional areas in cell m, $Q_{y}$ denotes the weight of each of the 13 functional areas or, in the case of mixed functional areas, the average value of the two single functional areas, and $C_{m}$ denotes the travel vitality index of the functional areas in cell m.

### 3.4. Evaluation of the Levels of Service in Parks and Squares

According to the Urban Green Space Planning Standards of the Ministry of Housing and Urban—Rural Development of the People’s Republic of China, the service radius of parks and squares should be graded according to type and scale. The size of the service radius is closely related to the scale of the population served and the scale of the park itself. The larger the service radius of a park, the larger the population it serves and the larger the park’s area.

In our study, service coverage differed from the traditional concept of park coverage in that it referred to the proportion of the total area covered by a park’s service areas to the total area of the park. The formula was as follows:(5)$$C=\frac{\sum PA}{A}\times 100\%$$
where C denotes the service coverage, ∑PA denotes the sum of the service areas excluding all overlaps, and A denotes the total area of the park.

The service overlap ratio referred to the proportion of each park’s service areas that overlapped with other parks to the sum of the service areas of all parks. The formula was as follows:(6)$$O=\frac{\sum CO-\sum PA}{\sum CO}\times 100\%$$
where O is the service overlap ratio, ∑PA is the sum of the service areas excluding all overlaps, and ∑CO is the sum of the service areas of each park.

### 3.5. Attractiveness Assessment of Parks and Squares

The vitality of parks and squares is closely related to their attractiveness to urban residents. To understand the attractiveness of various parks and squares to urban residents more accurately, references were made to the classification of urban leisure tourism resources. The acquired data from parks and squares were divided into four categories, each of which contained several subcategories that corresponded to scenic spots (238) within 13 functional areas. According to the standard “classification, investigation, and evaluation of tourism resources” [56], the magnitude of the attractiveness of the four types of parks and squares was evaluated using on-site investigations and expert scoring. Weights were assigned and divided into four grades based on 1. Then, the weight of the importance and attractiveness of each type of park and square to urban residents was obtained (Table 3).

### 3.6. Weight Determination and Data Processing

In this study, the attractiveness of parks and squares was taken as an important factor in measuring their vitality. At the same time, traffic network density, population density and the service levels of parks and squares were also taken into account to determine the scoring criteria for each index that affected vitality, and the overall weight was set to 1. Based on the results of our field survey, 20 experts with planning backgrounds and familiarity with Jiaozuo were consulted, and existing studies were referenced [52,57,58]. The result of these consultations was that the attractiveness of parks and squares was assigned a weight of 0.3, the travel vitality index of urban residents was assigned a weight of 0.3, the service levels of parks and squares was assigned a weight of 0.2, and the urban functional area density was assigned a weight of 0.2. In the process of our comprehensive evaluation of the vitality of parks and squares in Jiaozuo, the value ranges were quite different and not directly comparable because of the different natures of the indices. To facilitate the subsequent analysis and calculations, we adopted the deviation standardization method to normalize the evaluation indices for the vitality of parks and squares to eliminate dimensional differences. The formula was as follows:(7)$$U_{ij}=\frac{X_{ij}-\min\left\{X_{j}\right\}}{\max\left\{X_{j}\right\}-\min\left\{X_{j}\right\}}$$
where i denotes the study unit (i = 1, 2, …, n), j denotes the evaluation index (j = 1, 2, …, n), $X_{ij}$ and $U_{ij}$ are the original and normalized values of the index j for the study unit i, respectively, and $\min\left\{X_{j}\right\}$ and $\max\left\{X_{j}\right\}$ are the minimum and maximum values of the original value of the index j, respectively.

## 4. Results and Analysis

### 4.1. Classification and Identification of Urban Functional Areas in the Main Urban Areas of Jiaozuo

The types of urban functional areas were mainly determined by the types and numbers of POIs. The main urban area in Jiaozuo was divided into 594 functional area units of 1 km^2^. The number of POIs in each unit was weighted using the partition identification method, which not only considered the spatial distribution of POIs within the whole study area but also considered their density in the functional area units; therefore, we could identify urban functional areas more accurately. Using the above method, 13 functional areas and their corresponding $C_{i}$ value ranges were identified in the main urban area in Jiaozuo, as shown in Figure 5. Since each functional area corresponded to a different $C_{i}$ range, this range was used as the basis for the identification of functional areas. The results showed that high-density areas had overlapping functions and strong overall recognition. Most functions were centered around Jiefang District and Shanyang District and showed dispersing east–west clustering characteristics.

Using Equations (1) and (2) to calculate the values of $F_{i}$ and $C_{i}$, the spatial distributions of single functional areas, mixed functional areas and data-free areas in the four districts of the main urban area in Jiaozuo were obtained (Figure 6). In Figure 5, it can be seen that the single functional areas and mixed functional areas generally showed spatial distribution patterns of single functional areas surrounding mixed functional areas. The mixed functional areas were mostly located in the core of the main urban area and showed clustered distributions, while the single functional areas were mostly located in the peripheral areas and showed fragmented distributions. More specifically, the single functional areas and the data-free areas were mainly distributed in the north of Zhongzhan District, the north of Jiefang District, the southwest of Shanyang District and the northeast of Macun District. The mixed functional areas were concentrated in the east of Zhongzhan District, the central and southern parts of Jiefang District, northern central Shanyang District and the southwest of Macun District. Using the statistics for the three types of functional areas, 230 mixed functional units were located in the central area of the main city, and 192 single functional units were found, mainly scattered around the mixed functional units in the peripheries of the city. There were 173 data-free units, most of which were distributed around single functional units at the urban boundaries.

Our overlay analysis of the different functional units showed the spatial distribution patterns of 594 specific single functional units and mixed functional units (Figure 7). The single functional areas were determined by the $C_{i}$ value of a certain type of data within the functional unit reaching 50% or above. Combined with the number of single functional areas and their proportions, as shown in Figure 8, it was found that company enterprises, governmental organizations and social groups and scenic spots accounted for 29.69%, 20.83% and 19.27%, respectively. Then, sports, leisure and transportation facilities, accommodation services and financial insurance services accounted for smaller proportions. Companies and governmental agencies occupied larger areas, and it was relatively easy for them to form single functional units. Scenic spots were considered as public service areas due to their services provided and their overall dispersion, local concentration and distribution within the main urban area. Commercial single functional spaces, such as restaurants and shops, were more sporadically distributed in Zhongzhan District and Macun District, where there were fewer mixed functional areas. Residential single functional units, such as accommodation services and commercial housing, were distributed in Shanyang District (one of the older urban areas), which also had the lowest proportion of single functional units (1.04%). In general, Shanyang District had more single functional areas (70), and the functional categories were diverse. Macun and Zhongzhan Districts had single functional areas with similar functional types and quantities, and Jiefang District had the smallest number of single functional areas (15).

If the $C_{i}$ value of each area in the functional units did not exceed 50%, the two types of areas with the largest proportions were selected according to the $C_{i}$ value to reflect the nature of the urban mixed functional areas. Mixed functional areas were mainly distributed in the core of the main urban area (Figure 7), which was the main area for urban functions. Combined with the number and proportion of mixed functional areas, as shown in Figure 9, it was found that the mixed functional areas were mainly distributed in Shanyang District and Jiefang District as mixed government–finance and government–companies areas, accounting for 5.22% and 4.78% of the areas, respectively. Mixed shops–companies and transportation–finance areas followed, accounting for 3.91% of the areas. The proportion of mixed catering–lifestyle functional areas was the lowest, at 0.43%. It could be seen that governmental organizations and companies had the most single functional units and mixed functional units. The mixed functional areas formed of commercial and residential units (such as catering, commercial housing, etc.) were distributed in the central areas of the mixed functional areas, which helped to create a rich shopping experience for urban residents. The spatial layout characteristics of these areas were very different from those in single functional areas. From the final results of our superposition analysis of the functional units in the main urban area in Jiaozuo, it could be seen that the central area of the main urban area had a high degree of functional mixing. Combined with the spatial population distribution map (Figure 3), it can be seen that the degree of functional mixing in urban areas and the population density showed similar distribution characteristics. According to the principle that functional areas should serve urban residents, it could be inferred that the urban spatial structures were relatively reasonable.

### 4.2. Calculation of the Travel Vitality Index of Urban Residents

The overall vitality levels of parks and squares were determined using the travel vitality index of urban residents in their areas and the service levels of the parks and squares. One of the most important factors that affects the ability of urban residents to travel is road network density, which is positively related to the travel vitality index of residents, i.e., the higher the road network density in a functional unit, the higher the travel vitality index. The normalization and superposition of the population and road network densities were analyzed according to the constructed POI saliency model [59,60]. The public awareness index was found to be an important factor affecting POI weighting (Table 4). By weighting the public awareness of different functional areas, a model for the travel vitality index of urban residents was established to obtain more detailed spatial distributions of the travel vitality index of residents in the urban functional units (Figure 10).

Using the natural discontinuity point method proposed in ArcGIS to visualize the travel vitality index of residents for 594 functional units, we found that the spatial distributions of the travel vitality index of urban residents in different areas had obvious concentric ring radiation characteristics and significant spatial differences. In terms of regional location, Jiefang District had a high mobility index, as it was very busy and had more prosperous areas located in its center. Shanyang District had more generally prosperous areas, but it was less busy and had fewer prosperous areas. Zhongzhan District contained more cool areas, and Macun District was the area with the lowest mobility index. The most prosperous areas included Tianhe Park, Rule of Law and Culture Square, People’s Park and Zhuque Park, which showed a contiguous distribution around the railway station. The busier areas, which mostly contained places such as churches and universities, were mostly distributed around other very busy areas. The generally prosperous areas were characterized by certain clustering scales on the whole and were more often located in the middle of the adjacent Jiefang and Zhongzhan Districts. Cooler areas were the norm on the peripheries of the main urban areas, such as the northeastern part of the Macun District and the northern part of the Zhongzhan District. These areas covered more data-free functional areas, mostly in the primary industry with a low population density, underdeveloped transportation networks and low accessibility.

### 4.3. Evaluation of the Levels of Service of Parks and Squares

Based on the service radius standards for the various types of parks stipulated in the Urban Green Space Planning Standards (Table 5), the service coverage of 238 parks and squares in the main urban area was calculated, and the service coverage rate and service overlap rate of the parks and squares in the four districts were also obtained. The overall service levels of parks and squares in the four districts were quantified by considering the two aforementioned indicators. While service coverage could better reflect the service scope of the parks and squares, the service overlap rate exposed their uneven spatial distributions. By comparing the service coverage rates and service overlap rates, the overall service levels of the parks and squares in the four districts of the main urban area in Jiaozuo were obtained (Table 6). Our analysis showed that Jiefang District had the highest service coverage rate, while the other three districts had significantly less service coverage, especially Macun District. Among them, only the coverage rate of parks and squares in Jiefang District was higher than the overlap rate, indicating that the distribution of parks and squares in Jiefang District was relatively balanced. Shanyang District, Zhongzhan District and Macun District all had higher service overlap rates than coverage rates, showing their dense distributions of parks and squares. Among them, the area of parks and squares in Shanyang District was 174.20 km^2^, but the service coverage rate was only 40.38%, and the service overlap rate was as high as 52.09%, indicating that the spatial distribution of parks and squares in Shanyang District was unbalanced. Zhongzhan District ranked second because the service coverage rate was the lowest, reflecting the unbalanced spatial distribution of its parks and squares. The number of parks and squares in this district was found to be insufficient and could not meet the needs of urban residents, so there is much room for improvement.

### 4.4. Analysis of Park and Square Vitality

In this study, the attractiveness of parks and squares, the richness of their surrounding businesses, the road network and population densities and the service levels of the area were used as indicators to measure the vitality of parks and squares. The weights of the influencing factors were determined from existing research and expert scoring. Using the data presented in Table 3, Equation (4) and Table 6, as well as the weights of the influencing factors, the vitality of 238 parks and squares was calculated, and the indicators were standardized and normalized. According to the categories of “higher vitality”, “high vitality”, “medium vitality”, “low vitality” and “lower vitality”, a vitality classification map of the 238 parks and squares in the main urban area in Jiaozuo was obtained using the natural discontinuity method (Figure 11). By overlaying this map onto the map of the travel vitality index of urban residents, we found that most functional units showed an overlap between high- and low-vitality parks and squares and high and low travel vitality indices of urban residents. However, a few functional units with high travel vitality indices of urban residents did not contain parks and squares, indicating that as a medium-sized city, Jiaozuo should further increase the number of parks and squares to meet the needs of its residents. Overall, the vitality levels of the parks and squares in the Jiefang District were central to their attractiveness, and a total of 24 of the highest-vitality parks and squares were distributed in the Jiefang District. People’s Square, People’s Park, South Tower, etc., were representative of these parks, as the service levels were more abundant, the road network was more developed, and the flow of people was more attractive. The parks and squares with medium to high vitality were mostly distributed around the middle of Shanyang District, the western part of Macun District and the eastern part of Zhongzhan District, such as Taiji Square, Longyuan Lake Park, Macun Park and the Ecological Botanical Gardens. Due to the lag in urban construction, the surrounding functional areas mostly had single functions, resulting in the relatively low vitality of the parks and squares. The parks and squares with lower vitality were sporadically distributed in the peripheral areas of Macun District and Zhongzhan District. The surrounding road networks were sparse and residents’ accessibility to the parks and squares was low, so their vitality was greatly affected.

In addition to the different service levels of the parks and squares, we studied the impact of the functional mixing, road network density, spatial population distribution and attractiveness of parks and squares on their vitality in each district. Under the condition of the same service level, the main urban area was disassembled according to the administrative districts, and the current vitality of the parks and squares in each district was analyzed. All parks and squares within each district were ranked according to decreasing attractiveness to obtain the final results (Figure 12). It can be seen from the figure that the parks and squares with the highest vitality among the four districts were in the Jiefang District (3.5 points), followed by those in the Shanyang District, and the parks and squares with the lowest vitality were in the middle station area (1.2 points). The vitality of the parks and squares in the four districts showed a high degree of correlation with the service levels of the areas. We found that in Jiefang District (i.e., the district with the highest vitality score), approximately 50% of the parks and squares had high levels of attractiveness, the number of parks and squares in this area was more than adequate, and the spatial layout was more reasonable. This indicated that there was a high potential for residents to visit parks and squares in this area and that the overall vitality of parks and squares was high (Figure 12a). Nearly 60% of the parks and squares in Shanyang District had high attractiveness levels, which exceeded the proportion in Jiefang District; however, due to the relatively low levels of business richness and road network accessibility, combined with the higher service overlap rate than coverage rate, the spatial distribution was uneven, resulting in lower overall vitality of parks and squares in the district than in Jiefang District (Figure 12b). There were large variations in the vitality of parks and squares in Zhongzhan District, and there were areas where the vitality value was 0 (Figure 12c). On the one hand, while the area of Zhongzhan District is 1.75 times larger than that of Jiefang District, we found that the number of parks and squares was only 32.3% of that in Jiefang District, indicating that the parks and squares within the district did not meet the needs of its residents. On the other hand, this could be due to the poor correlation between the distribution of parks and squares with high levels of attractiveness and areas with high travel vitality indices of residents. The overall vitality of parks and squares in Macun District was low (Figure 12d) due to its marginal location in the eastern part of the main urban area, as well as the single functional areas and poor traffic accessibility brought about by lagging urban development, which affected the vitality of the parks and squares. By looking at the vitality of the parks and squares in the main urban area, it can be seen that the smaller and more densely populated Jiefang District had a higher number and better laid out parks and squares. The other three districts, which were relatively sparsely populated, covered large areas and contained good natural environments. In future urban construction or green space development, parks and squares could be used to alleviate population pressures and offer great development potential that is more conducive to improving the overall spatial vitality of the main urban area.

## 5. Conclusions and Discussion

On the basis of our original research into the division of urban functional areas, this paper introduced our research into the vitality of urban parks and squares. By comparing the differences in construction levels between large and medium-sized cities in China, physical environmental factors, such as urban functional mixing, traffic network density, spatial population distribution, the number of parks and squares and their layout maturity, were extracted as the factors that most affected the vitality of urban parks and squares. From the perspective of urban functional areas, multisource data were used to identify single or mixed functional areas. The travel vitality index of urban residents was also calculated, along with the service levels of the parks and squares. Overall, four indicators were selected to evaluate the vitality of parks and squares in medium-sized cities. The results showed the following: (1) according to the method of using the $C_{i}$ value to identify single functional units, the single functional units in the four districts of the main urban area in Jiaozuo were mainly distributed in the central parts of Jiefang District and Shanyang District, the eastern part of Zhongzhan District and the western part of Macun District; (2) the functional mixing in the main urban area in Jiaozuo generally presented the spatial distribution characteristics of high density in the center and lower surrounding densities, although the functional mixing in the four districts was quite different, and the Jiefang and Shanyang Districts contained approximately 64.13% of all mixed functional units; (3) the travel vitality of urban residents in the four districts was affected by the functional mixing, road network density and spatial population distribution, forming the spatial characteristics of concentric circular radiation; (4) by analyzing the service levels of the parks and squares in each area and weighting the service coverage and service overlap rates, we found that the service levels of the parks and squares in Jiefang District were the highest and the spatial distribution was the most reasonable (followed by Shanyang District), whereas the parks and squares in Zhongzhan District had the lowest service levels and spatial distribution; (5) from our comprehensive evaluation of the four indicators, the order of the vitality of the parks and squares in the four districts was Jiefang District > Shanyang District > Macun District > Zhongzhan District; (6) to study the influence of functional mixing, road network density, spatial population distribution and the attractiveness of parks and squares on their vitality in the four districts, 238 parks and squares were studied, and under the condition of the same service level, the vitality of the parks and squares in each area was compared. The order of vitality of parks and squares from high to low was Jiefang District (1.0–3.5), Shanyang District (0.2–2), Zhongzhan District (0–1.2) and Macun District (0–1.4). We also found that the vitality of the parks and squares was closely related to the urban functional mixing and the travel vitality index of urban residents.

From the above conclusions, it can be seen that urban functional mixing, road network density, population density, facility coverage and attractiveness are important factors that affect the vitality of parks and squares. Therefore, we propose the following suggestions. (1) The formation of vibrant parks and squares should be promoted by improving urban functional mixing. Some scholars have analyzed the significant impacts of water, facility coverage and surrounding population density on the vitality of parks and squares from both supply and demand perspectives. It has been proven that increasing the functional mixing in the areas around parks and squares can improve their vitality [61]. The higher the utilization rates of mixed functional areas, the higher the vitality of urban spaces [62]. In the future, planning and construction processes in medium-sized cities should consider the spatial distribution and functional area configuration of parks and squares to give full play to the rational layout of functional areas and promote the formation of high-vitality spaces. (2) Accessibility is the most important factor affecting the vitality of spaces [63], and dense road network structures help to improve the accessibility of parks and squares, thus forming vibrant parks and squares [64]. Given that the road networks in medium-sized cities are not yet perfect, the demand for roads around service spaces should be fully considered in future road planning. Ideal road networks help to stimulate the potential of service spaces, such as parks and squares. (3) In areas with a high population density, the number of parks and squares should be appropriately increased to improve the service levels of those areas. As the main users of parks and squares, urban residents, through their participation, would help to maintain the high vitality of parks and squares. Using the elastic model of variable importance, some scholars have shown that to increase spatial vitality, population density is the most important factor, followed by functional mixing and transportation networks [65]. Therefore, changes in population growth and spatial distributions should be taken into account in the medium- and long-term planning of cities under construction. On the one hand, public demand for urban functions should be met, but on the other hand, the high-quality and sustainable development of cities should be ensured.

Based on the division of urban functional areas, we used multisource data, such as POI data, population data and road network data, to analyze the vitality of microspaces in medium-sized cities by partition identification. Our results could provide a new perspective for the spatial distribution and vitality assessment of parks and squares in medium-sized cities, as well as new ideas for subsequent improvements in the vitality of parks and squares. However, this study was based on the mature traditional model. Some data assignment processes, such as POI data, have a certain subjectivity. When determining the travel vitality index of urban residents, some of the data we used, such as the WorldPop population data, had a certain time lag and could not reflect real-time population distributions. Therefore, in a follow-up study, newer data that can reflect the spatial and temporal population distributions will be considered to analyze the travel vitality index of urban residents. Furthermore, a dynamic analysis of the use of parks and squares in terms of time and space will be carried out to evaluate the real-time vitality of parks and squares to provide further reference for the planning and construction of future service spaces in medium-sized cities.

## Figures and Tables

**Figure 1 ijerph-19-15238-f001:**
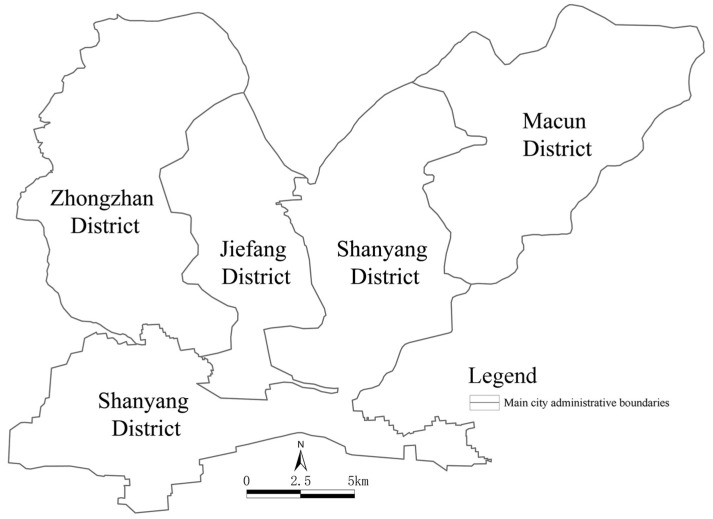
The main urban area in Jiaozuo, consisting of four districts.

**Figure 2 ijerph-19-15238-f002:**
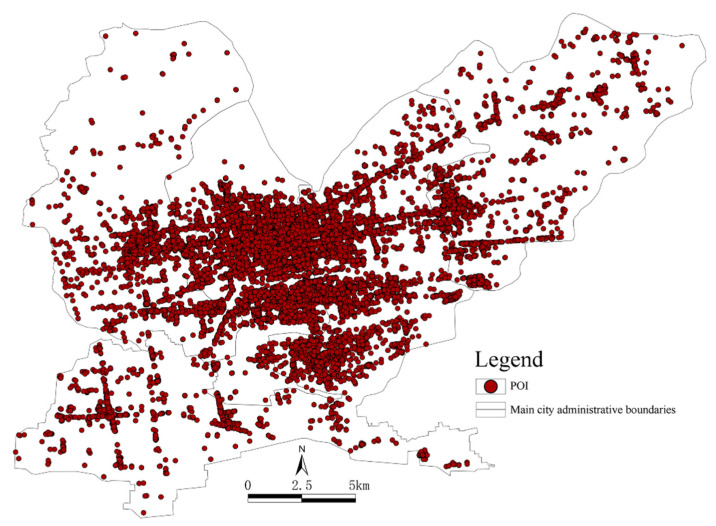
The distribution of POI data from the main urban area in Jiaozuo.

**Figure 3 ijerph-19-15238-f003:**
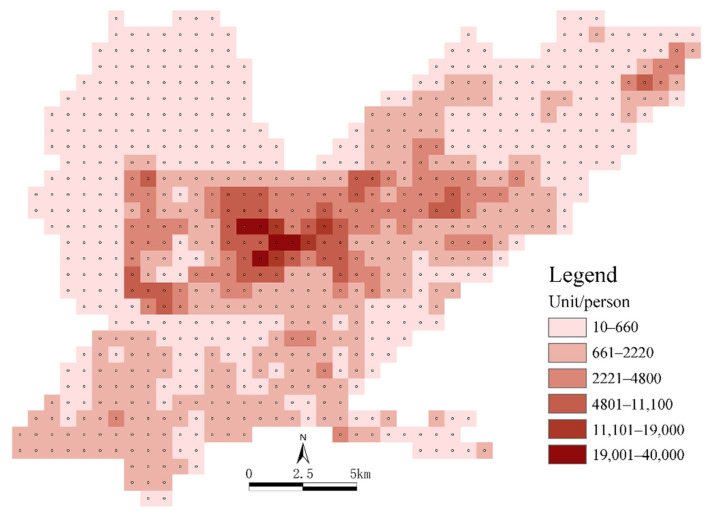
The population density of a 1 km × 1 km grid in the main urban area in Jiaozuo.

**Figure 4 ijerph-19-15238-f004:**
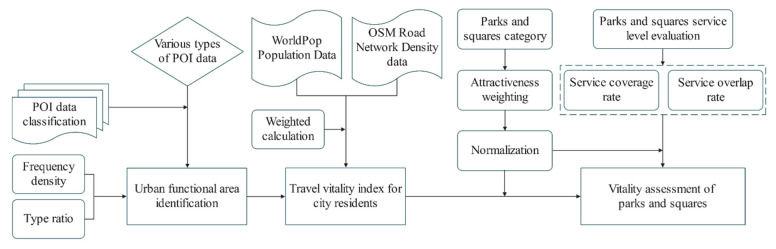
The framework for our study on park and square vitality.

**Figure 5 ijerph-19-15238-f005:**
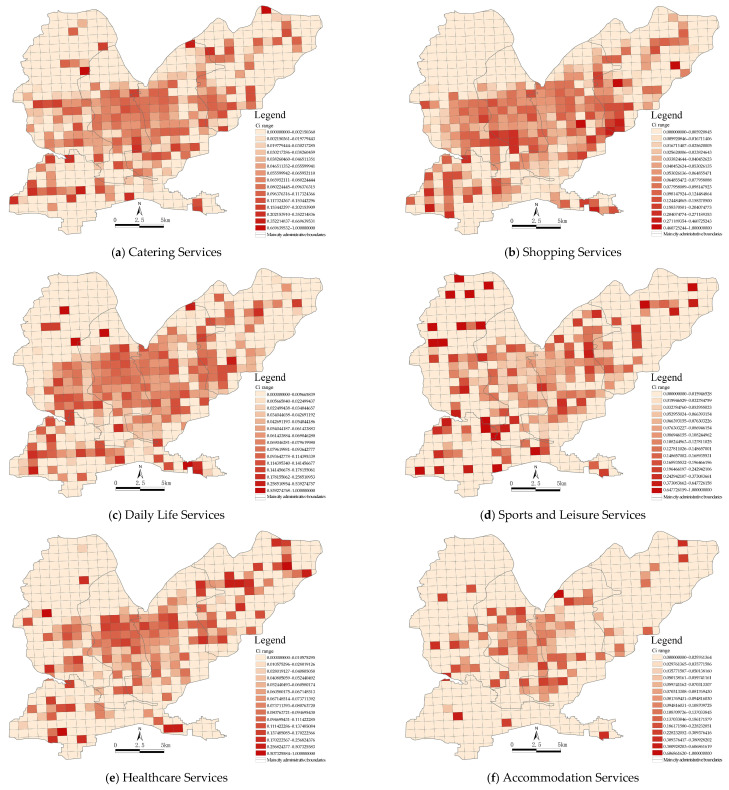
A functional classification identification map of the main urban area in Jiaozuo based on POI data ($C_{i}$).

**Figure 6 ijerph-19-15238-f006:**
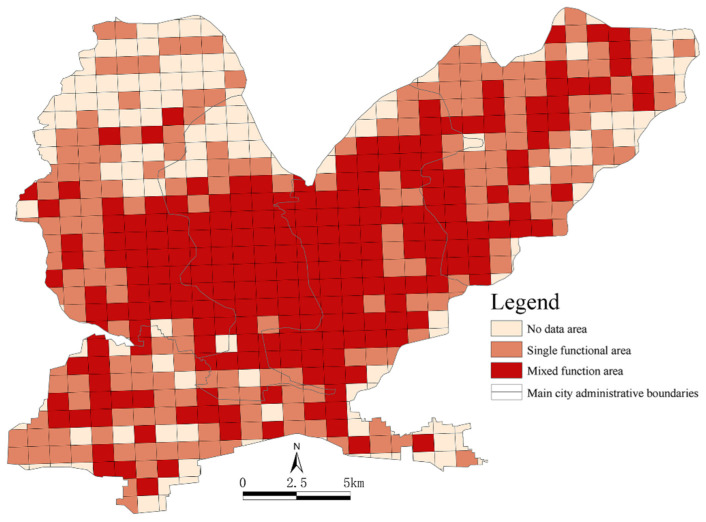
The spatial distributions of the urban functional units in the main urban area in Jiaozuo.

**Figure 7 ijerph-19-15238-f007:**
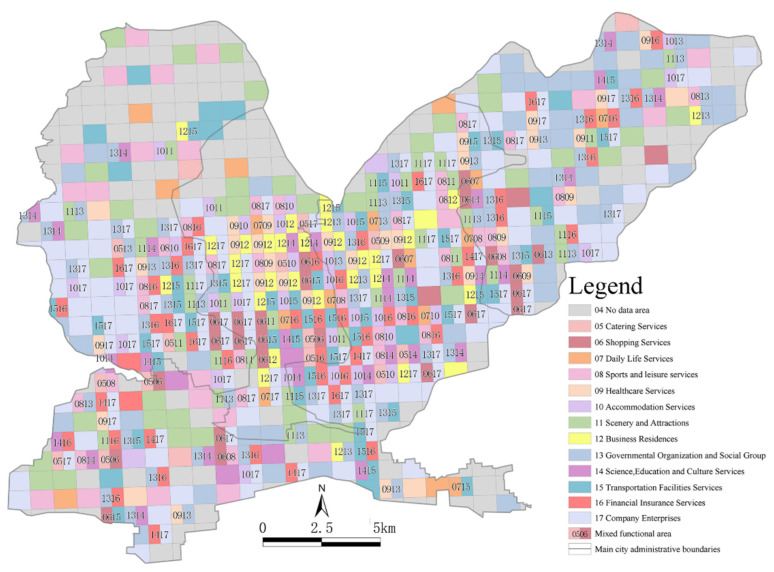
The distribution of urban functional units in the main urban area in Jiaozuo.

**Figure 8 ijerph-19-15238-f008:**
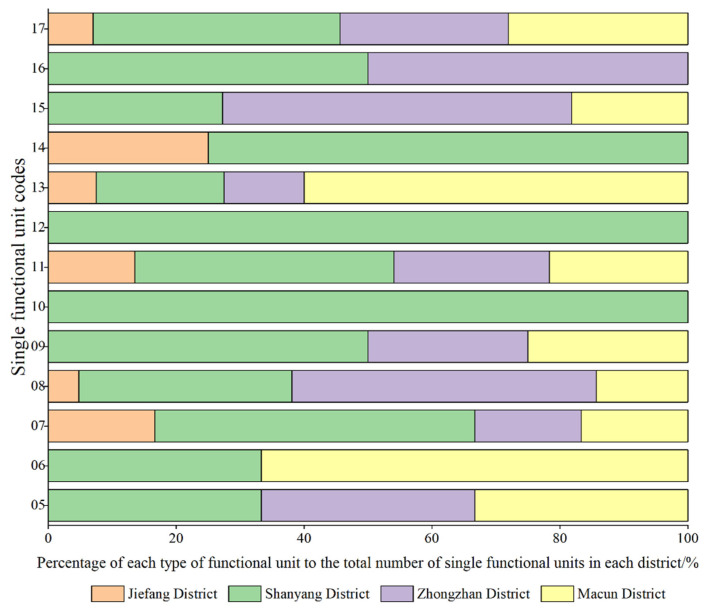
An analysis of the identification results for the single functional units in the main urban area in Jiaozuo.

**Figure 9 ijerph-19-15238-f009:**
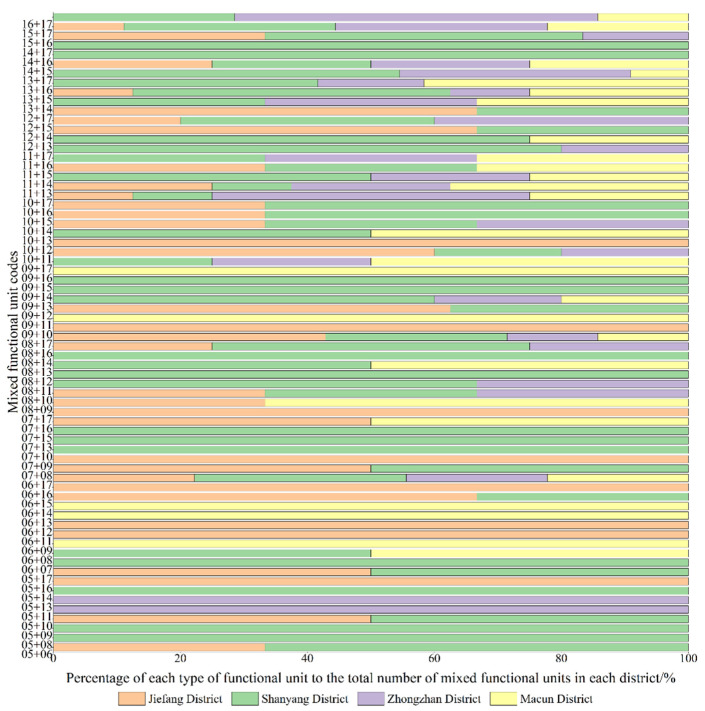
An analysis of the identification results for the mixed functional units in the main urban area in Jiaozuo.

**Figure 10 ijerph-19-15238-f010:**
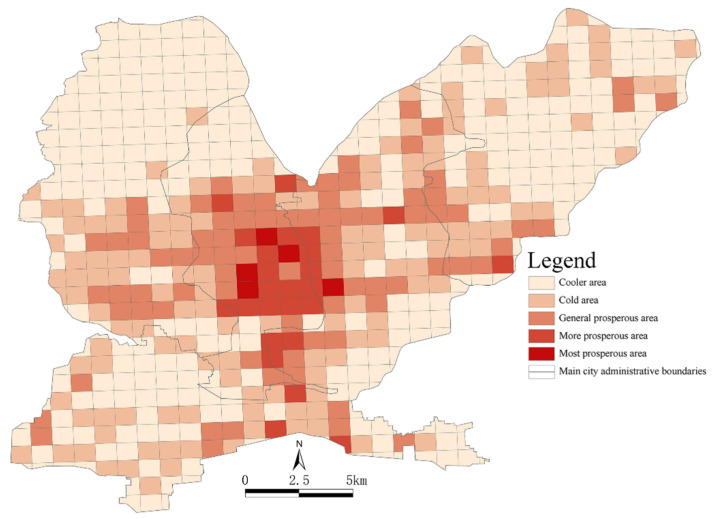
The travel vitality index of urban residents in the main urban area in Jiaozuo.

**Figure 11 ijerph-19-15238-f011:**
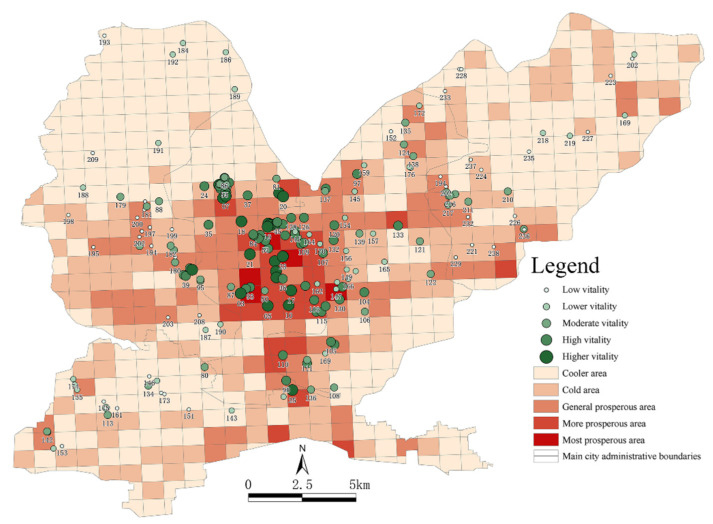
The vitality of parks and squares in the four districts of the main urban area in Jiaozuo.

**Figure 12 ijerph-19-15238-f012:**
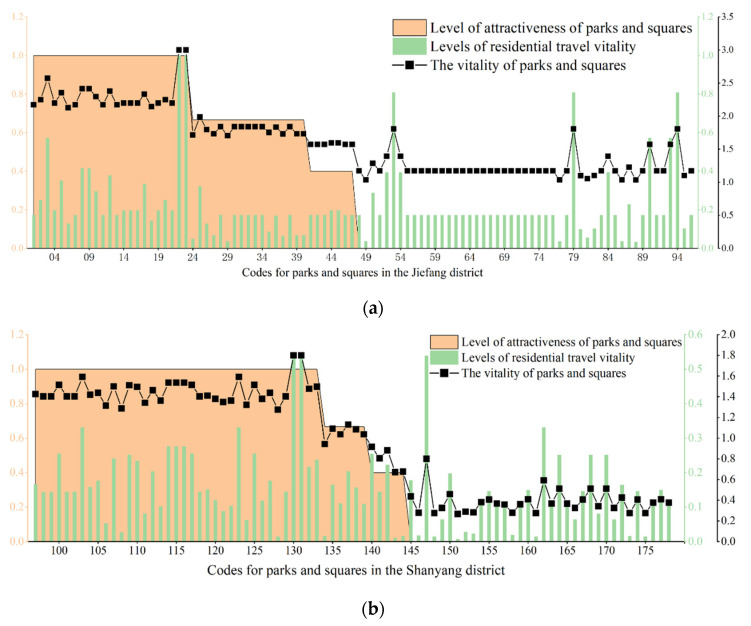
The vitality of parks and squares in the four districts of the main urban area in Jiaozuo: (**a**) the vitality of parks and squares in Jiefang District; (**b**) the vitality of parks and squares in Shanyang District; (**c**) the vitality of parks and squares in Zhongzhan District; (**d**) the vitality of parks and squares in Macun District.

**Table 1 ijerph-19-15238-t001:** The classification of POI data from the main urban area in Jiaozuo.

Code	Category	POI Contents
05	Catering Services	Chinese restaurants, foreign restaurants, fast food restaurants, tea houses, casual dining places, pastry stores, dessert stores, coffee shops, cold drink stores
06	Shopping Services	Shopping malls, supermarkets, convenience stores, specialty stores, buildings, home appliance and electronic stores, home building material markets, clothing, shoe, hat, and leather goods stores
07	Daily Life Services	Beauty salons, photography and printing stores, laundromats, telecommunications offices, maintenance sites, post offices, moving companies, bathing and massage places, baby service places
08	Sports and Leisure Services	Entertainment venues, sports and leisure service venues, leisure venues, vacation and recreation venues, sports venues, theaters, golf-related
09	Healthcare Services	Medical and healthcare sales stores, clinics, general hospitals, specialty hospitals, healthcare services, emergency centers, veterinary clinics, disease prevention centers
10	Accommodation Services	Hotel guest houses, hotels, accommodation services-related
11	Scenery and Attractions	Park square, scenic spot, scenic spot-related
12	Business Residences	Residential areas, business residential-related, industrial parks, buildings
13	Governmental Organization and Social Group	Government agencies, social groups, traffic vehicle management, government, and social groups-related, public prosecution and law agencies, industrial and commercial tax agencies, democratic parties, foreign agencies
14	Science, Education, and Culture Services	Schools, museums, science and technology museums, scientific and educational and cultural places, training institutions, literary and artistic groups, driving schools, media organizations, cultural palaces, exhibition centers, exhibition halls, libraries, planetariums, archives
15	Transportation Facilities and Services	Train stations, coach stations, bus stations, parking lots
16	Financial Insurance Services	Banks, ATMs, financial and insurance service providers, securities companies, insurance companies, finance companies, bank-related
17	Company Enterprises	Companies, corporate enterprises, factories, well-known enterprises, agriculture, forestry, and fishery bases

**Table 2 ijerph-19-15238-t002:** Basic information on the study data from 2020.

Data Name	Data Form	Data Type	Volume of Data	Year
POI/pc	Point	Vector data	41,902	2020
Parks and squares/pc	Point	Vector data	238	2020
WorldPop/pc	Grid	Raster data	721	2020
OpenStreetMap/strip	Line	Vector data	1216	2020

**Table 3 ijerph-19-15238-t003:** The classification of parks and squares and their attractiveness weights.

Major Categories	Minor Categories	Attractiveness Weighting
Public Recreation	City squares, parks, public recreation buildings	1
Nature, General	Natural scenic spots, nature reserves, forest parks, geological parks, wetland parks	0.75
Cultural Venues	Memorial halls, cultural halls, museums, aquariums	0.55
Historic Sites	Ruins, ancient buildings, ancient gardens, tombs and mausoleums, places of worship and religious activities	0.25

**Table 4 ijerph-19-15238-t004:** The public awareness of POIs.

Category	Public Awareness	Category	Public Awareness
05. Catering Services	55.62	12. Business Residences	30.57
06. Shopping Services	81.46	13. Governmental Organizations and Social Groups	35.50
07. Daily Life Services	81.46	14. Science, Education, and Culture Services	67.06
08. Sports and Leisure Services	50.10	15. Transportation Facilities and Services	100.00
09. Healthcare Services	50.69	16. Financial Insurance Services	30.57
10. Accommodation Services	55.62	17. Company Enterprises	30.57
11. Scenery and Attractions	82.45		

**Table 5 ijerph-19-15238-t005:** The standard service scopes for city parks of different sizes.

Type		Park Size (hm^2^)	Size of Population Served (10,000 People)	Service Radius (m)
Integrated Parks		≥50.0	>50.0	>3000
		20.0–50.0	20.0–50.0	2000–3000
		10.0–20.0	10.0–20.0	1200–2000
Residential Park	Community Park	5.0–10.0	5.0–10.0	800–1000
		1.0–5.0	1.5–2.5	500
	Garden Tour	0.4–1.0	0.5–1.2	300
		0.2–0.4	—	300

Note: In the old city, parks and squares of 0.2–0.4 hm^2^ were used to calculate the service radius coverage to the nearest 300 m, and the historical and cultural districts were adjusted downward to 0.1 hm^2^.

**Table 6 ijerph-19-15238-t006:** The service coverage and service overlap indices of parks and squares in the main urban area in Jiaozuo.

Urban Areas	District Administrative Area (km^2^)	Service Coverage Rate C (%)	Service Overlap Rate O (%)	Relative Service Level
Jiefang District	71.71	80.25%	50.19%	Higher
Shanyang District	174.20	40.38%	52.09%	General
Zhongzhan District	125.00	26.50%	52.40%	Lower
Macun District	118.90	28.60%	50.71%	Lower

## Data Availability

The study did not report any data.

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
