# Peer review of "Research on the Vitality Evaluation of Parks and Squares in Medium-Sized Chinese Cities from the Perspective of Urban Functional Areas"

_ijerph, 2022, doi:10.3390/ijerph192215238_

Round 1

Reviewer 1 Report

The paper has an appropriate literature review. The paper contributes to the body of the literature and has added value because the authors used a new relevant methodology a new idea for the subsequent activation of the vitality of park squares with multisource data. The analyzed theme is relevant to this journal yet challenging. I recommend an English minor proofing of the text. 

Reviewer 2 Report

This manuscript evaluates the vitality of parks and squares in the city of Jiaozuo, China. Studying vitality in combination with the urban functional area is an interesting topic per se. However, it seems that the manuscript is not ready for publication at this stage because of the following concerns. Therefore, I recommend Major Revision to see what the manuscript looks like after completion and reach a decision.

1) I think the manuscript requires showing its connection with the scope of IJERPH. The manuscript needs to explain the implications of vitality on public health and environmental research. The manuscript in its current form is more compatible with the journals covering urban and spatial planning matters rather than the journals in the area of Public, Environmental, and Occupational Health.

2) In the introduction, the research problem needs to be clearly stated and straight. However, the problem statement is a bit ambiguous. For example, in the first paragraphs, the importance of vitality is explained, and the reader expects that the manuscript will offer new vitality indicators and evaluation framework. But, in the subsequent paragraphs (e.g., lines 57-61), the manuscript talks about how vitality can be improved in medium-sized cities. This way, the reader becomes confused about whether vitality measurement is the problem or vitality improvement is the research objective or both.

3) The process of selecting vitality indicators is unclear, and the reviewed literature for the identification and classification of such indicators is not sufficient.

4) The way the weights of the indicators are obtained is not explained.

5) The research is case-specific, and consequently, cannot help readers understand new facts about the concept of vitality in medium-sized cities. In other words, studying one case is not enough to lead the research outputs to certain remarks.

6) Applied methods are not new, and the manuscript does not contribute to the existing body of knowledge methodologically as well.

7) Innovations of the manuscript are vague. I think the innovations are minor, as the results are not generalizable, applied methods are common, and the research problem is not well-justified.

8) Legends of the maps are not readable.

Reviewer 3 Report

I have read the paper titled “Research on the Vitality Evaluation of Parks and Squares in Chinese Medium-sized Cities from the Perspective of Urban Functional Areas”; and this manuscript presents useful and important findings which add to the growing understanding of the potentials of the urban public spaces that meet the people’s demand for acquiring a high-quality urban life, and also it highlights the importance of adopting the targeted strategies that improve the spatial vitality and create high-vigor parks and squares in medium-sized cities in the future.

However, I have made some comments, as they have written down, and minor revisions are needed I would encourage the authors to address these and resubmit as this is important work for publications.

Whilst the content is clear, the English language needs to be improved throughout, the manuscript would benefit from a thorough proof read by a native English speaker to improve grammar.

§  The abstract should have the research problem and the argument. The abstract must represent the whole paper, and it stands alone.

§  It is suggested to present the structure of the research at the end of the introduction.

§  The novelty of the work relative to previously published works should be emphasized in the abstract, introduction, results and conclusions parts.

§  The adopted model, methods and measures, is not the optimal model thus it is better to upgrade the paper with “research limitations”, described what has not been covered in this research.

Round 2

Reviewer 2 Report

The authors did a great job and addressed most of my comments, and I think the manuscript is ready for publication at this stage.

As the last point, I highly recommend the authors provide a separate paragraph in Section 1 and describe the contributions in detail after explaining gaps and problems. I know the readers can perceive innovations from Section 1, but stating them clearly and directly in a different paragraph can reinforce the paper's quality.
